# Current Status and Future Prospects of Biolubricants: Properties and Applications

**Rakesh Narayana Sarma** [1,2] and **Ravikrishnan Vinu** [1,2,*]

1 Department of Chemical Engineering, Indian Institute of Technology Madras, Chennai 600036, India; nrakeshenergy@yahoo.com
2 National Center for Combustion Research and Development, Indian Institute of Technology Madras, Chennai 600036, India
* Correspondence: vinu@iitm.ac.in; Tel.: +91-44-2257-4187

**Abstract:** Biolubricants generated from biomass and other wastes can reduce the carbon footprint of manufacturing processes and power generation. In this paper, the properties and uses of biolubricants have been compared thoroughly with conventional mineral-based lubricants. The biolubricants, which are currently based on vegetable oils, are discussed in terms of their physicochemical and thermophysical properties, stability, and biodegradability. This mini-review points out the main features of the existing biolubricants, and puts forward the case of using sustainable biolubricants, which can be generated from agro-residues via thermochemical processes. The properties, applications, and limitations of non-edible oils and waste-derived oils, such as bio-oil from pyrolysis and bio-crude from hydrothermal liquefaction, are discussed in the context of biolubricants. While the existing studies on biolubricants have mostly focused on the use of vegetable oils and some non-edible oils, there is a need to shift to waste-derived oils, which is highlighted in this paper. This perspective compares the key properties of conventional oils with different oils derived from renewable resources and wastes. In the authors' opinion, the use of waste-derived oils is a potential future option to address the problem of the waste management and supply of biolubricant for various applications including machining, milling applications, biological applications, engine oils, and compressor oils. In order to achieve this, significant research needs to be conducted to evaluate salient properties such as viscosity, flash point, biodegradability, thermo-oxidative and storage stability of the oils, technoeconomics, and sustainability, which are highlighted in this review.

**Keywords:** biolubricant; base stock; biodegradability; vegetable oil; pyrolysis bio-oil; hydrothermal liquefaction bio-crude

## 1. Introduction

Lubrication is defined as the process or technique by which the wear of one or both moving surfaces in close proximity is reduced by using a substance called lubricant in between the surfaces. Lubricant carries or helps to carry the pressure generated (or load) between the opposing surfaces [1]. Lubricant acts as anti-friction media, facilitates smooth operation, maintains reliable machine functions, and decreases the risk of frequent failure [2]. A lubricant may be a liquid, a semi-solid (grease), or a solid (including coatings and particles) [3]. The major objectives of lubrication include: (a) reducing wear and preventing heat loss due to the contact of moving surfaces; (b) protecting the surface from corrosion by reducing the oxidation; (c) acting as an insulator in transformer applications; and (d) acting as a sealant against dust, dirt and water. While it is difficult to eliminate wear and heat totally by using lubricants, they can be minimized and controlled to acceptable levels [1]. Minami [3] pointed out the three main functions of lubricants as follows: (a) controlling friction; (b) cleaning contact; and (c) cooling the contact.

The world depends on fossil-fuel based products, such as petroleum products and petrochemicals, to a great extent in the industrial and transportation sectors. These products can cause serious environmental hazards and pollution. Biomass and solid wastes can be potential candidates for generating fuels for power generation, such as [4–6] liquid fuels, methanol [7], bio-oil [8,9], and platform chemicals [10]. In the field of lubrication, which is an important application of petroleum products, mineral-based lubricants can be substituted by biolubricants that can potentially address some of the related environmental effects. Biolubricants exhibit superior lubrication properties over conventional lubricants, in addition to being renewable and biodegradable [11]. Biolubricants usually have their origin from vegetable oils, plant polymeric carbohydrates, and wax esters [12]. Biolubricant preparation involves a transesterification reaction wherein an ester is reacted with an alcohol to produce another ester through interchange of the alkyl group. The transesterification of vegetable oils generates fatty acid alkyl esters of different alcohol chain lengths. The reaction is usually catalyzed by mineral acids and bases, and the final product, i.e., fatty acid alkyl esters, can be utilized as fuel, biodiesel, and lubricating agents [1].

The basic constituents of a lubricant include base oil and additives, which enhance the properties of the oil. The typical ratio of base oil and additives in the lubricant is 90:10. Synthetic liquids such as hydrogenated polyolefins, esters, silicones, and fluorocarbons are used as base oils in conventional lubricants, while vegetable oils are used in biolubricants. Because of renewability and environmental considerations, vegetable oils have also received attention as base oils [13]. The major constituents of vegetable oils include triacylglycerols (98%), diglycerols (0.5%), free fatty acids (0.1%), sterols (0.3%), and tocopherols (0.1%) [14]. The triglyceride structure is composed of three hydroxyl groups esterified with the carboxyl groups of fatty acids. Owing to the high molecular weight, the triglycerides possess high viscosity and viscosity index.

The term biolubricant is used to represent all lubricants that are easily biodegradable and non-toxic to human beings and the environment. While its use is still very limited as compared to mineral oil-based lubricants [15], they are promising candidates as they are renewable and emit net zero greenhouse gases to the environment [2]. Some of the key terms to be understood include mineral oil, base stock, and base oil [16]. Mineral oil is the liquid hydrocarbon by-product obtained from crude oil distillation. Base stock is usually produced by a single manufacturer to the same specification independent of the feedstock or manufacturer's location (API 1509 2005). Base oil refers to a single type of base stock or a blend of multiple base stocks used to prepare a lubricant.

It has been described that biolubricant has excellent lubricity and causes a reduction in friction coefficient, frictional forces and wear compared to other lubricants [17]. As vegetable oils inherently have excellent tribological properties, they are considered to be very effective lubricants. However, the hydrolytic stability and thermal stability of the vegetable oils have been reported to be poorer than conventional mineral oils, and should be improved [13]. The economic and environmental significance of base oils and sustainable lubricants was discussed by Shah et al. [18]. As nearly 66% of fuel energy is lost to the surroundings due to thermal, frictional, transmission and other components, the design of sustainable biolubricants and tribological advancements are imperative to enhance fuel efficiency. Estimates suggest that more than 1% savings in GDP can be achieved annually by implementing better lubricants in manufacturing, transportation, power generation, and residential sectors. Another interesting topic of research is the chemical modification of vegetable oils to produce an alternative to petroleum-based materials [19]. Biolubricants can also be prepared from waste cooking oil and cyclic oxygenates through a four-step catalytic process [20]. It has been pointed out by some researchers that viscosity is the most important property of lubricants, as it determines the amount of friction between two surfaces. To reduce wear, lubricants with higher viscosity result in a higher viscosity ratio and lower wear rates [21]. Eco-friendly multipurpose lubricating greases from vegetable residual oils have been studied, and it has been observed that they have superior tribological performance as compared to commercial grease [22]. By studying the rheological

and wetting behavior of Environmentally Acceptable Lubricants (EALs) for use in stern tube seals, it has been shown that the operational shear rate of the ship should be considered while selecting a stern tube lubricant [23]. Researchers produced environmentally friendly ethylene glycol di-esters (EGDEs) as biolubricants from various vegetable oils by applying CaO as a heterogeneous base catalyst through the transesterification of fatty acid methyl esters (FAMEs) and ethylene glycol (EG) [24]. Biolubricants can also be produced by the transesterification of rapeseed and castor oil methyl esters with various alcohols (2-ethyl-1-hexanol, 1-heptanol and 4-methyl-2-pentanol) using titanium isopropoxide as a catalyst [25].

The authors have noticed the need for an article which comprehensively describes various aspects of biolubricants (including source, preparation, properties, biodegradability, and application), and the need to develop alternate lubricants from non-edible and waste agroresidues. Even though biomass has been considered as a source of fuels and chemicals, the lubrication properties of biomass-based oils have not been consolidated in line with conventional lubricants. The present article thus aims to bridge the gap in the literature pertaining to biolubricants. The objectives of this mini review are four-fold: firstly, to describe the sources, preparation and greenness of biolubricants vis-à-vis conventional lubricants; secondly, to discuss the major properties of biolubricants including viscosity, thermo-oxidative stability, pour point, eco-toxicity, hydrolytic stability, and most importantly, biodegradability; thirdly, to present the development of biolubricants from non-vegetable oil, non-edible sources such as lignocellulosic agriresidues and wastes via thermochemical transformation, with the aim of determining the fit of pyrolysis bio-oil and hydrothermal liquefaction bio-crude as biolubricant base stocks; finally, to present the applications of biolubricants in different fields along with future prospects and research directions.

## 2. Sources and Preparation of Lubricating Oils

Conventional lubricants contain high molecular weight hydrocarbons derived from the vacuum residue in the refinery as base stocks. Biolubricants, which are so far vegetable oil-based, are constituted by unsaturated fatty acids as base stocks. Owing to oxidizable functional groups, they possess poor thermal stability and poor oxidation stability, which make them unsuitable for use in applications such as lubricating oils. Using chemical modification processes such as transesterification, epoxidation and hydrogenation, unsaturated fatty acid content can be reduced to make the vegetable oils suitable for use in engines, thereby making them comparable to traditional lubricants [26]. The life cycle of conventional mineral oil-based lubricant and biolubricant can be explained using Figures 1 and 2, respectively. Based on the literature review, the types of base stocks used for lubricating oils are summarized in Table 1. The American Petroleum Institute (API) has categorized the base oils into five groups (I–V) based on whether they are mineral oil-based or synthetic [27,28]. The classification also depends on the hydrocarbon composition of the oils and the sulfur content in them. Table 2 presents the major fatty acids that constitute the vegetable oil-based biolubricants.

Additives are added to base stocks to improve the physicochemical and thermophysical properties as well as the chemical and thermal stability of the final lubricant formulation. Lubricant additives are classified based on their role in lubrication systems, which include anti-wear or anti-oxidation agents and friction modifiers. They are also classified as per their working function and their working site. The first category includes tribo-improvers, rheo-improvers, and maintainers. Tribo-improvers improve the tribological performances of lubricants. Rheo-improvers are concerned with the fluidity of the base oil. Maintainers reduce the degradation of the substances participating in the lubrication system and keep the lubricants and the machine components in good condition. Based on the working mechanism, the additives can be classified as chemical additives if they undergo chemical reactions, or physical additives, if they work without any chemical changes [3].

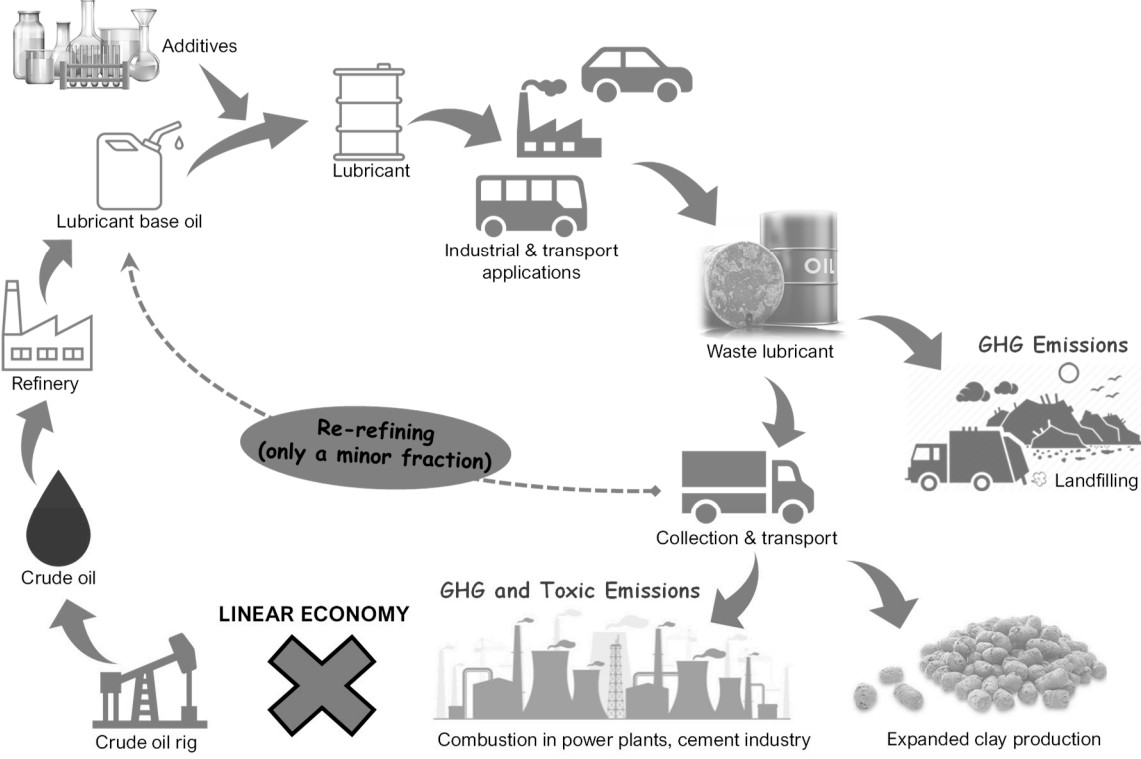

**Figure 1.** Life cycle of conventional lubricants.

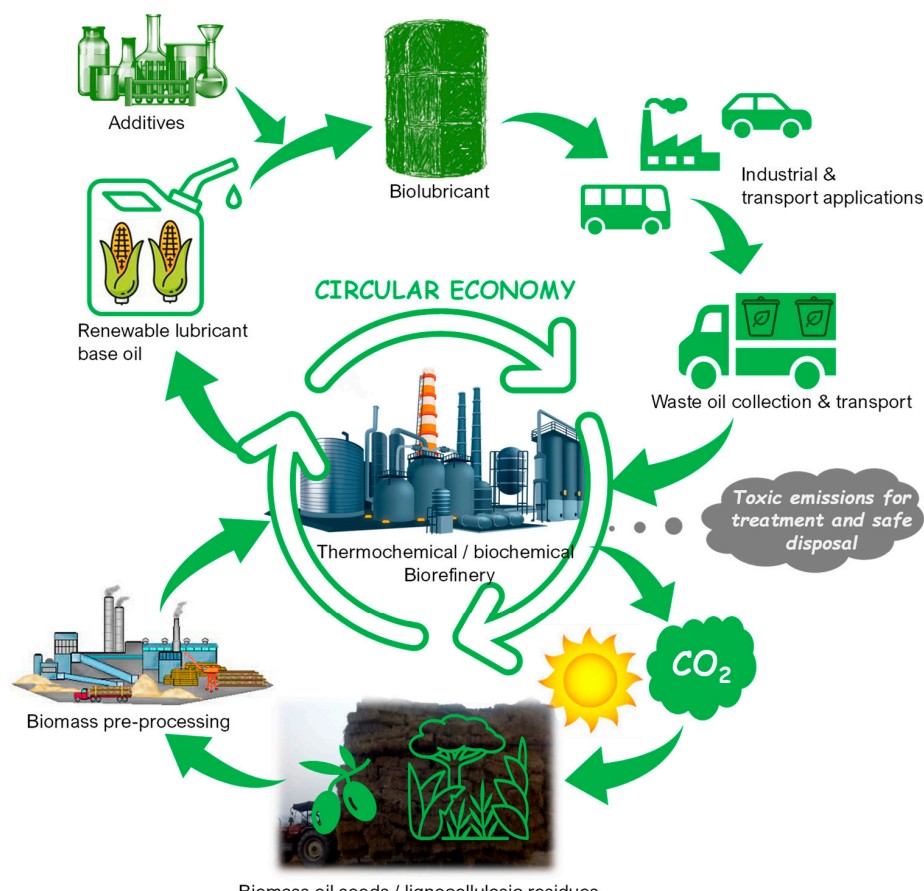

**Figure 2.** Life cycle of biolubricants.

**Table 1.** Base stocks for lubricant oils.

| Sl. No. | Base Stock | Source | Reference |
|:---:|:---:|:---:|:---:|
| 1 | Mineral oil base stocks | They are obtained from crude oil processing after solvent refining, catalytic dewaxing, hydrotreatment, and hydrocracking. They can be naphthenic, aromatic, or paraffinic in nature. They mainly include API Group I, II, and III oils. | [26–28] |
| 2 | Re-refined oil base stocks | These are obtained from refined petroleum products after the removal of volatile and insoluble components and contaminants via acid/clay treatment. They mainly include API Group I, II, and III oils. | [26] |
| 3 | Synthetic oil base stocks | They are obtained from petroleum crude oil after chemical modification via hydrotreating and hydroprocessing. They mainly include poly-$\alpha$-olefins, silicones, polyolesters, phosphate ester-based and polyalkylene glycol-based oils. They mainly include API Group IV and V oils. | [26] |
| 4 | Biomass base stocks | They include plants and animal-based oils including vegetable oils, lipids, and oils derived from agro-residues and wastes via thermochemical and catalytic processing. | [15,26,29] |

**Table 2.** Important fatty acids in plant-based oils [30,31].

| Common Name | Molecular Formula | Fatty Acid Type |
|:---:|:---:|:---:|
| Palmitic acid | $C_{16}H_{32}O_2$ | Saturated |
| Stearic acid | $C_{18}H_{36}O_2$ | Saturated |
| Oleic acid | $C_{18}H_{34}O_2$ | Monounsaturated |
| Linoleic acid | $C_{18}H_{32}O_2$ | Diunsaturated |
| Linolenic acid | $C_{18}H_{30}O_2$ | Triunsaturated |
| Ricinoleic acid | $C_{18}H_{34}O_3$ | Unsaturated fatty acid |

*Biodegradable Grease*

One effective way of providing lubrication to machine components is by using grease. Even though liquid lubricant flows easily, a reservoir is needed to store it. Solid lubricants need direct contact at the point of lubrication for effective lubrication. Grease is a common lubricant in rolling bearings, and comprises base oil, thickener, and small amounts of additives. Commercial greases are mostly produced from petrochemical base oils and thickeners [32]. A semi-solid grease can provide the benefits of liquid lubricants without needing a reservoir, and also the advantages of solid lubricants by maintaining body structure. In applications such as the wheel bearing of an automobile, in which excessive heat is produced, liquid lubricants tend to thin down and could leak out of the bearing seals. Lubricating grease is a colloidal dispersion of a thickening agent dispersed in a matrix of lubricant base fluid, and owes its consistency to a gel-forming network. As cooling and cleaning functions need fluidity, solid lubricants and greases may be applied to machine elements where contamination and localized heat generation due to friction

are not serious factors. Researchers have discussed biodegradable greases from palm oil industry wastes [33] and castor oil [34]. It has been pointed out that environmentally safe and biodegradable organic compounds can be used to produce biodegradable grease [35]. Sánchez et al. [36] presented formulations based on various types of acylated chitosan to produce stable gel-like dispersions in castor oil that are suitable for application as biodegradable grease.

Lubricating greases are a group of lubricants that exhibit gel-like characteristics. The gel-like behavior is imparted by the thickening agent, which is usually metallic soap, phyllosilicate, or polyurea compounds [37]. These are two-phase colloidal suspensions comprising mineral oil and a thickener forming a three-dimensional gelling network. Fatty acid soaps of calcium, lithium, aluminum, sodium and barium are usually utilized as thickeners [38]. Borrero-López et al. [39] pointed out that the residual lignin-containing fractions resulting from hydrolysis and kraft pulping biomass conversion processes may be utilized as thickening agents. The bio-based oleogels with suitable lubricating properties can be prepared by chemical functionalization, while their chemical structure and composition can modulate the functional properties of them.

Some researchers have focused on the use of castor oil for formulating biodegradable grease. Sánchez et al. [40] described the tribological characterization of green lubricating greases formulated with castor oil and various biogenic thickening agents. They noted that castor oil-based biodegradable greases provide similar or lower values of the friction coefficient than traditional lithium greases, which was found to depend on the nature of the thickening agent and the tribological contact. Gallego et al. [41] subjected many lignocellulosic pulps from different sources to cross-linking with hexamethylene diisocyanate, and dispersed the formulation in castor oil to obtain gel-like semisolid lubricants. Acar et al. [42] discussed that biodegradable lubricating greases can be prepared using high-oleic sunflower oil and castor oil, biodegradable thickening agents such as natural cellulose fibers of different chain lengths, and glyceryl and sorbitan stearates. Cortés-Triviño et al. [43] pointed out that epoxy-modified cellulose pulp-based biodegradable greases can provide excellent thermal stability. Modifying the pulp with an epoxide compound strongly altered the friction coefficient and wear relative to the use of castor oil alone as a biolubricant in specific low-speed regimes.

### 3. Properties of Biolubricants

The major functions of biolubricants, i.e., reducing friction and wear, the dispersion of deposits, the inhibition of rust/corrosion, the dissipation of heat, and the sealing of critical contact joints, are reflected in the properties of the base oil. The base oil is expected to possess optimum viscosity and viscosity index, low volatility, low deposit formation, low temperature solidification, good hydrolytic and thermo-oxidative stability and biodegradability [13]. A good biolubricant should have its boiling point distribution towards high temperatures, high viscosity index, corrosion prevention capability, high thermal stability, low freezing point, and high anti-oxidation potential [2]. While these are the preferred characteristics of any biolubricant, specific applications do demand unique properties. For example: (a) engine oil application demands low emissions of volatile organic compounds and polyaromatic hydrocarbons; (b) metal working fluids require good emulsifiability; (c) hydraulic oils require low compressibility and a quick release rate of air; (d) transmission oils and gear oils require high weld load; and (e) greases should possess good anti-wear and anti-scoring properties. The salient properties are explained in the following sections.

#### 3.1. Viscosity

Viscosity is a key parameter that determines the time to replace the lubricant in a device. The viscosity depends on factors such as the concentration of paraffins and the additives that affect the internal friction among the molecules, and it increases with the increase in the chain length of the hydrocarbon portion of the fatty acid or alcohol in ester-based biolubricants. Owing to increased hydrogen bonding interactions, the

presence of hydroxyl groups in the lubricant formulation or the addition of polyols modifies the viscosity index [15,44,45]. The viscosity index (VI), a metric used by lubricant users and refiners, describes the effect of temperature changes on the viscosity of the oil [46]. VI measures the temperature dependency of viscosity [47]. A higher value of VI indicates that the temperature will not affect the viscosity to any large degree and vice versa [46]. The reference values of VI are determined according to the standard methods described by the American Society for Testing and Materials (ASTM) D2270-10 [48] and of the Brazilian Association of Technical Standards (ABNT NBR) 14,358 [49].

In order to understand the temperature effects on the kinematic viscosity of base stocks and lubricants, VI was first proposed by Dean and Davis of Standard Oil in 1929 [50]. Viscosity and VI can be measured using ASTM D445-97 and D2270-93. ASTM D445 includes the Standard Test Method for the Kinematic Viscosity of Transparent and Opaque Liquids. The measurement method includes tracking the flow time of transparent or opaque liquids through a calibrated glass capillary. Typically, API Group I and Group II base oils exhibit VI in the range of 80–120, while Group III base oils exhibit VI greater than 120. It is important to note that Group I base oils contain <90 wt.% saturated hydrocarbons and >0.03 wt.% sulfur, while Group II and III oils contain >90 wt.% saturated hydrocarbons with <0.03 wt.% sulfur in them. Poly-$\alpha$-olefins (Group IV base oils) contain >99 wt.% saturated hydrocarbons, and they also exhibit a very high VI > 120 [27,28]. From Table 3, it is evident that the kinematic viscosity values of the hydrocarbon-based mineral oils are high (40–150 cSt), while those derived from the degradation of polyethylene or polypropylene are even higher (200–700 cSt). High molecular weight polymers when degraded at low temperatures, or for insufficient periods of time, tend to form waxy compounds with carbon chain lengths > 30, which tends to increase the viscosity. However, after catalytic upgradation, the viscosity can be brought down to the range of typical oils extracted from edible and non-edible seeds. The formation of aromatic hydrocarbons such as benzene, toluene, ethylbenzene, xylene (BTEX) and other alkyl benzenes during the catalytic treatment using acidic catalysts such as zeolites also tends to decrease the viscosity of the oil. Pyrolysis bio-oil derived from biomass also possesses low viscosity, but due to its oxygen-rich composition, it cannot be used as a biolubricant. The co-pyrolysis of polymers/plastics with biomass followed by catalytic treatment is also shown to lead to the low viscosity of oil, which is typically in the range of vegetable oil biolubricants.

### 3.2. Thermo-Oxidative Stability

Auto-oxidation of the lubricant is promoted due to localized high temperatures caused by frictional heat generated by the rubbing of solid surfaces against each other. This tends to alter the viscosity of the lubricant due to reactions that promote the cleavage of long chain hydrocarbon molecules. Therefore, due to exothermic oxidation and endothermic pyrolysis reactions, the thermo-oxidative stability of the lubricant decreases with usage [51]. Thermo-oxidative stability can be tested using the RPVOT test (ASTM D2272). This involves the evaluation of the oxidation stability of the biolubricants in the presence of a copper catalyst and water at 423 K and 620 kPa of oxygen.

Jedrzejczyk et al. [52] showed that lignin-based additives can be used for improving the thermo-oxidative stability of biolubricants. They tested four different lignins—commercial Protobind P1000 soda lignin from straw, solvolytically fractionated Protobind P1000 lignin, and two lignin fractions from a reductively catalyzed fractionation (RCF) of native birch wood—in biolubricant formulations with castor oil as the base oil. They reported that the lignin fractions exhibited excellent performance in comparison to the butylated hydroxytoluene (BHT), a petroleum-based antioxidant, utilised commonly as an antioxidant. The formulations of modified lignin in castor oil held better thermo-oxidative stability, as illustrated by their increased oxidation induction time. In addition, rheological and tribological tests demonstrated similar, or in some cases, improved lubricating properties in comparison to castor oil.

Thermal and thermo-oxidative stability can also be assessed by analyzing the change in apparent activation energy of the oil with conversion using thermogravimetric analysis. Tripathi and Vinu [53] evaluated the thermal stability of synthetic and semi-synthetic oils that were aged at 120, 149 and 200 °C. Fourier transform infrared spectroscopy coupled with chemometric models can be used to determine the Total Acid Number (TAN), Total Base Number (TBN), oxidation index, nitration index, and sulfation index of the biolubricant at different thermo-oxidative simulated ageing conditions.

### 3.3. Pour Point

Pour point is the temperature below which the lubricant loses its flowability. In biolubricants, it is related directly to the viscosity index. The presence of ternary alcohols, such as trimethylolpropane (TMP), reduces the pour point of the biolubricant, even though it tends to reduce the thermo-oxidative stability of the lubricant [15]. It can be measured using ASTM D5949, which involves the determination of pour point by applying a burst of nitrogen gas into the lubricant sample, while simultaneously cooling it. It is usually performed in an automated instrument, which also detects movement of the surface of the test sample using an optical device.

### 3.4. Ecotoxicity

Ecotoxicity is a metric used to characterize the environmental toxicity of a lubricant formulation. It is a major property that determines whether a lubricant formulation can irreversibly affect living things. As aqueous ecosystems are prone to damage by the organic and hydrocarbon components of lubricants, it is vital to determine lubricant water toxicity. This is defined as the potential of a lubricant to poison target organisms such as bacteria, algae, small fish, or laboratory rats. ASTM D6081-20 is the standard protocol to test the aquatic toxicity of the lubricant.

### 3.5. Hydrolytic Stability

The resistance of biolubricants to chemical attack, especially when water molecules are involved either as a reactant or a product, is characterized by hydrolytic stability. The ASTM D2619-21 standard test method is used to determine the hydrolytic stability of petroleum or synthetic-based hydraulic fluids. Biolubricants for use as insulation fluids require high water solubility and a high dielectric constant. Therefore, the assessment of hydrolytic stability becomes imperative.

### 3.6. Biodegradability

If a lubricant can be structurally decomposed by enzymes or microorganisms through an aerobic or anaerobic process, it can be considered biodegradable [54]. A lubricant can be considered biodegradable if the percent degradation in a standard test exceeds a certain value. Vegetable oils are more biodegradable than mineral-based oils [2]. The environmental concern around the depletion of mineral reserves has sparked interest in biolubricants derived from natural triglycerides, as well as the fatty acids derived from them [55]. Biodegradability of the base fluid (or any other component) of an environmentally acceptable lubricant (EAL) depends on both its molecular properties and the test method utilised [56]. Importantly, the chemical composition of base oils can change during the application of lubricants, i.e., when they are subjected to varying temperature, air, humidity, metals, and pressure.

Standard methods for the determination of biodegradability have been discussed by Luna et al. [57]. CEC L-33-T-82 was one of the first methods used to evaluate biodegradability. In this approach, a standard fluid of known biodegradability is used. Both the standard and the sample are inoculated with microorganisms, and the reaction progress is monitored over a 28-day period. The quantification of carbon dioxide released from decomposition can be used to quantify biodegradation reactions, as is adopted in the ASTM D5864-11 method (Standard Test Method for Determining Aerobic Biodegradation of Lubricants and

Their Components). In this test, a natural aqueous environment is simulated, and the $CO_2$ generation is quantified.

The method presented by the Organization for Economic Cooperation and Development (OECD) is also based on $CO_2$ generation during the biodegradation of a sample. $CO_2$ is captured in a sodium or barium hydroxide solution, which is then titrated to quantify the $CO_2$ emission. A biodegradable sample is expected to exhibit more than 60% degradation over a 28-day period. For measuring the ultimate stage of the aerobic biodegradation, the OECD 301 B method is used in aqueous or soil medium as an ultimate biodegradation test [58]. In the Bartha Respirometer Method, the $CO_2$ generated is captured by a KOH solution [59,60]. The one global regulation defining environmentally acceptable biolubricants is the Vessel General Permit (VGP), which prescribes the discharge limit of oils into water bodies of vessels of different sizes. While better base stocks and additives are required for performance improvement, low aquatic toxicity and low bioaccumulation are the key properties of a good biolubricant. The ASTM D5864 method is based on the exposure of the lubricant to an inoculum under controlled laboratory conditions to determine the degree of aerobic aquatic biodegradation. ASTM D6731-18 involves the determination of the aerobic and aquatic biodegradability of lubricants using a closed respirometer. The biodegradability of greases essentially reflects the biodegradability of their base stocks [22]. Researchers have also pointed out that the high biodegradability of vegetable oils can make the vegetable oil-based greases as suitable alternatives to conventional greases [61].

**Table 3.** Physicochemical properties of some oils.

| Sl. No. | Oil | Density at 298 K (kg/m$^3$) * | Kinematic Viscosity at 313 K (cSt) | Oxidation Stability, 383 K, h | Cloud Point (K) | Flash Point (K) | Ref. |
|---|---|---|---|---|---|---|---|
| | | | **Non-edible oils** | | | | |
| 1 | Karanja | 918 | 4.80 | 6.0 | 282 | 423 | |
| 2 | Castor | 898 | 15.25 | 1.2 | 259.5 | 533 | |
| 3 | Neem | 885 | 5.20 | 7.2 | 287.5 | 317 | |
| 4 | Jatropha | 878 | 4.82 | 2.3 | 275.75 | 409 | |
| 5 | Tobacco | 887 | 4.25 | 0.8 | NR | 439 | |
| 6 | Mahua | 850 | 3.40 | NR | NR | 483 | |
| 7 | Rubber seed oil | 870.9 (at 313 K) | 31.4 | NR | NR | NR | |
| | | | **Edible oils** | | | | [2,15,55,62–71] |
| 8 | Coconut | 805 | 2.75 | 35.4 | 273 | 598 | |
| 9 | Sunflower | 878 | 4.45 | 0.9 | 276.42 | 525 | |
| 10 | Linseed | 890 | 3.74 | 0.2 | 269.2 | 451 | |
| 11 | Soybean | 885 | 4.05 | 2.1 | 274 | 598 | |
| 12 | Peanut | 882 | 4.92 | 2.1 | 278 | 450 | |
| 13 | Olive | 892 | 4.52 | 3.4 | NR | 591 | |
| 14 | Rice bran | 886 | 4.95 | 0.5 | 273.3 | 591 | |
| 15 | Rape seed | 880 | 4.45 | 7.5 | 269.7 | 525 | |
| 16 | Palm | 875 | 5.72 | 4.0 | 286 | 438 | |
| | | | **Other oils** | | | | |
| 17 | HTL biocrude | 940–960 | 110–350 | NR | 278 | 366 | [72,73] |
| 18 | Waste cooking oil | 908–955 | 35.3 | NR | 272 | NR | [74,75] |
| | | | **Pyrolysis-derived oils** | | | | |
| 19 | Biomass | 1100–1300 | 13–80 (at 323 K) | NR | NR | 323–373 | [76,77] |
| 20 | PS | 1100 | 1.4 | NR | NR | 375 | [78] |
| 21 | PP | 980 | 212 | NR | NR | 357 | [78] |

**Table 3.** *Cont.*

| Sl. No. | Oil | Density at 298 K (kg/m³) * | Kinematic Viscosity at 313 K (cSt) | Oxidation Stability, 383 K, h | Cloud Point (K) | Flash Point (K) | Ref. |
|---|---|---|---|---|---|---|---|
| 22 | Catalytically upgraded oil from PS | 979 | 1.63 | NR | NR | 356 | [78] |
| 23 | Catalytically upgraded oil from PP | 853 | 5.98 | NR | NR | 350 | [78] |
| 24 | LDPE | 856 | 476.6 | NR | NR | NR | [79] |
| 25 | PI | 841 | 6.4 | NR | NR | NR | [79] |
| 26 | PS–biomass mixtures | 1096–1192 | 2.0–2.75 | NR | NR | NR | [79] |
| 27 | PP–biomass mixtures | 615–942 | 681–729 | NR | NR | NR | [79] |
| 28 | LDPE–biomass mixtures | 832–867 | 139–187.5 | NR | NR | NR | [79] |
| 29 | PI–biomass mixtures | 880–892 | 4.1–7.5 | NR | NR | NR | [79] |
| 30 | Waste tire | 900 | 1.9 | NR | NR | 300 | [80] |
| 31 | Rice straw | 777–847 | 34.7–39.6 | NR | NR | 387–390 | [81] |
| 32 | Bagasse | 813–893 | 28.8–31.2 | NR | NR | 382–385 | [81] |
| **Synthetic and mineral oil** | | | | | | | |
| 33 | SAE20W40 | NR | 105 | NR | NR | 473 | [15,82] |
| 34 | Neat mineral oil | 880 | 62.9 | NR | NR | 497 | [83] |
| 35 | ISO VG32 | NR | >28.8 | NR | NR | 477 | [15] |
| 36 | ISO VG46 | NR | >41.4 | NR | NR | 493 | [15] |
| 37 | ISO VG68 | NR | >61.4 | NR | NR | 499 | [15] |
| 38 | ISO VG 100 | NR | >90 | NR | NR | 519 | [15] |
| 39 | R150 | NR | 150 | 15.52 | NR | 468 | [15] |

\*—Density measured according to ASTM D4052; NR—Not Reported; HTL—Hydrothermal liquefaction; PS—Polystyrene; PP—Polypropylene; LDPE—Low-density polyethylene; PI—Polyisoprene.

Biodegradability is a major requirement for current chemical industry products, and it is also vital from a circular economy viewpoint. The biodegradability metric is used to quantify the risk caused by the decomposition of products when they are deployed in the natural environment. Biodegradation, a natural process, occurs due to the action of microorganisms in the presence of oxygen, nitrogen, and minerals. Biodegradability depends on the quality and composition of the base oil used. Base-stock oils and finished lubricants are potentially exposed to the environment during their generation, distribution, service, and even during the disposal after the usage [57]. The biodegradability of biolubricants is presented in Figure 3. It is evident that vegetable oil-based hydraulic fluid, vegetable oil-based grease, and low erucic acid rapeseed have high values of ultimate biodegradation compared to the mineral oil-based hydraulic fluids and synthetic compounds. Biodegradability of some base stocks is also represented as percent loss at 21 days [29]. Typically, functional groups can be arranged in the following order based on the maximum percent loss at 21 days: alkyl benzenes (upto 20%) < polyethylene glycols (upto 75%) < aromatic esters (upto 90%) < polyols, diesters (upto 95%).

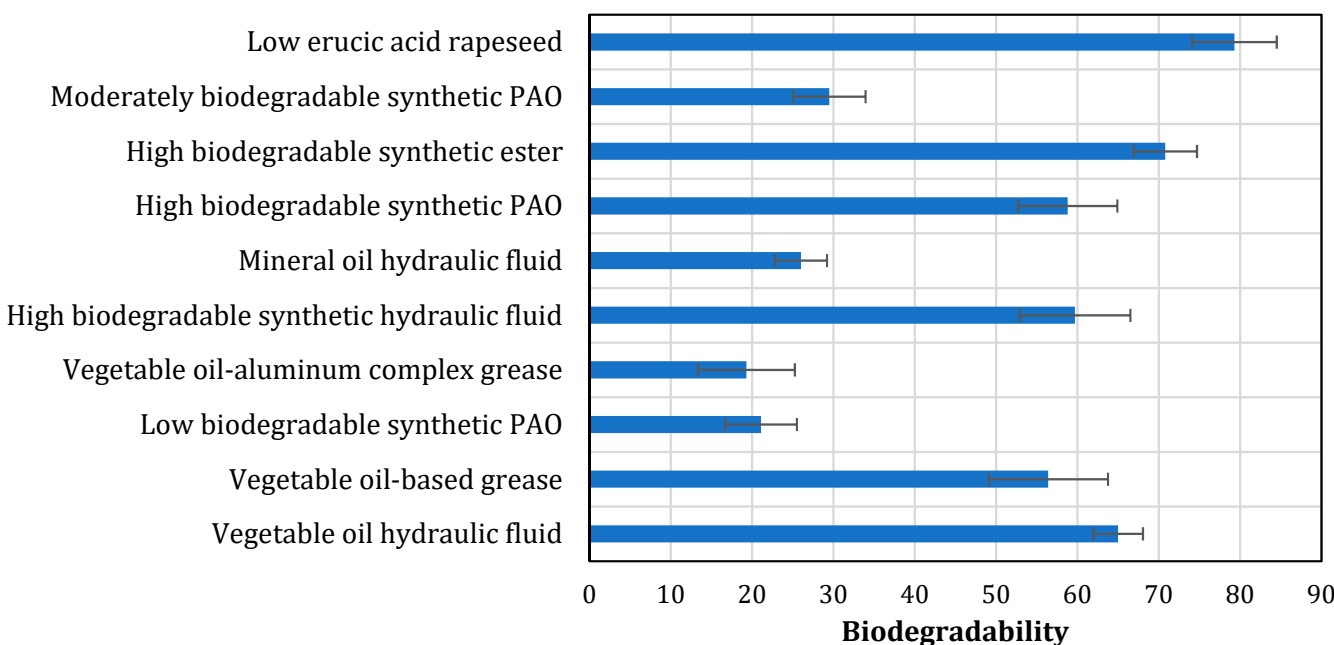

**Figure 3.** Biodegradability of different lubricant base stocks as per ASTM D5864.

## 4. Development of Biolubricants from Biomass via Thermochemical Techniques

Thermochemical conversion processes can be used to generate gaseous fuels [84] and liquid fuels from biomass [7,8]. Renewable fuels generated through thermochemical technologies such as pyrolysis, hydrothermal liquefaction (HTL) and the hydrodeoxygenation of bio-oils can potentially be cost competitive in comparison to other alternative fuel processes [85]. Lignocellulosic biomass is composed of three natural polymers—cellulose (30–60 wt%), hemicellulose (15–40 wt%), and lignin (15–30 wt%)—while algae species have proteins, lipids and carbohydrates, along with pigments such as chlorophyll, present in their matrix [86]. Liquid fuels can be generated from both lignocellulosic biomass agroresidues and lipid-rich microalgae by making use of pyrolysis and HTL process [87].

Pyrolysis refers to the thermal decomposition/cracking of organic matter in the absence of air. Based on the heating rate employed, thermal decomposition can occur on a scale of seconds to minutes. It is the primary chemical reaction, which is the precursor for both combustion and gasification processes. The thermal decomposition of biomass through pyrolysis results in the formation of liquid bio-oil, solid bio-char, and gaseous fractions rich in carbon dioxide, methane, hydrogen and carbon-monoxide [88–90]. Bio-oils are multi-component mixtures of different size molecules obtained from the fragmentation and depolymerization of lignin, cellulose, and hemicellulose. Typically, algal bio-oil is less dense than lignocellulosic bio-oil, while its viscosity falls in the typical range of wood-derived bio-oil. The presence of nitrogen-containing compounds such as indole, pyridine, amides, nitriles, and ammonia render the algal bio-oil pH alkaline (9.7), which is very different from that for lignocellulosic bio-oil (typically 2–3). Lignocellulosic biomass-derived bio-oil is a complex mixture of oxygenated organics including phenolics, alcohols, acids, dehydrated sugars, furan derivatives, carbonyl compounds, and aromatic hydrocarbons [91,92]. The acidic nature of lignocellulosic bio-oil is due to the presence of carboxylic acids and phenols. The major compounds in bio-oil from microalgae can be classified as aliphatic hydrocarbons, aromatic hydrocarbons (BTEX), nitrogenated compounds (including nitriles, amides and N-heterocyclic compounds), phenols, polycyclic aromatic hydrocarbons (PAHs), and others (such as alcohols, fatty acids, and esters) [93]. The elemental composition of bio-oil and petroleum crude oil are very different [77]. Owing to the decomposition of bio-chemical components in biomass, bio-oil is rich in oxygen, which imparts poor calorific value. More-

over, the oxygenated functional groups in bio-oil impart acidity and poor storage stability to the bio-oil.

The HTL process involves the high-pressure cooking of biomass or wastes in a hot, pressurized water environment for the necessary time to break down the biopolymeric structure into liquid bio-crude. The usual hydrothermal processing conditions are 523–647 K of temperature and an operating pressure ranging from 4 to 22 MPa. HTL is the only thermochemical process that is agnostic to the type of feedstock used and initial moisture content. As it is conducted in presence of water with low solid loading (10–20 wt.%), the use of wastewater or water with co-solvents offers opportunities to alter the product yields and their quality. The liquid oil from HTL is usually called bio-crude, and the other products include bio-char, aqueous phase, and gaseous fraction, rich in carbon dioxide. The high energy efficiency and low operating temperature in HTL makes the liquefaction process interesting and better than pyrolysis, which is more energy intensive [94].

Generally, HTL bio-crudes possess lower oxygen content and higher carbon content than pyrolysis bio-oils. This results in better calorific value of HTL bio-crude. The typical oxygen content in pyrolysis bio-oil from different feedstocks falls in the range of 30–50 wt.%, while that in HTL bio-crude is 10–30 wt.%. The elemental compositions of HTL bio-crudes from different feedstocks are well documented by Nallasivam et al. [95], and that of pyrolysis bio-oils are available in Gautam and Vinu [86]. The moisture content is also lower in HTL bio-crude than pyrolysis bio-oil. There are huge opportunities to tailor-make these fuels to produce water containing lubricants and cutting fluids. From Table 3, it is clear that HTL and pyrolysis-derived liquids can have comparable properties to that of conventional lubricants in terms of density, viscosity, and flash point, which makes them suitable for biolubricants, while at the same time solving the waste management issues and reducing the carbon footprint. The possibility of generating oils from wastes such as biomass, plastics, paper, refuse-derived fuels, and algae make these two technologies excellent candidates for producing eco-friendly lubricants for engineering applications. Additionally, the issues associated with the use of edible seeds for lubricant production are avoided.

## 5. Comparison of Biolubricants with Other Lubricants and the Need for Alternate Biolubricants

Conventionally, mineral oils have been used for the purpose of lubrication. However, as they are fossil-fuel based, their availability is an issue in the long run. Additionally, the disposal of mineral oils can cause pollution in aquatic as well as terrestrial ecosystems [2,96]. The burning of mineral oils as a lubricant can emit traces of metals, such as phosphorous, zinc, calcium, magnesium, and iron nanoparticles [2,97]. In this context, the use of environmentally friendly and benign biolubricants becomes imperative.

While vegetable oils can be used as lubricants, there are advantages and disadvantages [2]. It can be noted that vegetable oils have excellent lubricity in comparison to mineral oils. However, vegetable oils possess high VI, sometimes as high as 220, while a VI of 90–100 is normal for most mineral-based oils. Viscosity must be optimal, as higher viscosity affects the flowability of the lubricant. Another major property of vegetable oils is their high flash point, which relates to easy storability. Typically, the flash point of vegetable oils is ~600 K, while it is at least 25 K lower for mineral oils. While vegetable oils are renewable, biodegradable, usually less toxic, and reduce the dependency on petroleum oils, vegetable oils lack necessary oxidative stability in their natural form for lubricant application. Vegetable oils oxidize easily and become thick with a plastic-like consistency due to the polymerization of hydrocarbons at high temperatures. Moreover, vegetable oils possess an unpleasant smell, flushing propensity because of low viscosity, poor compatibility with paints and sealants, and a tendency to clog filters. Table 4 presents the salient advantages and disadvantages of biolubricants.

Waste cooking oil (WCO) can be a potential base stock for biolubricant preparation. In a study [98], WCO was chemically modified via epoxidation using $H_2O_2$ followed by

transesterification with methanol and branched alcohols (isooctanol, isotridecanol and isooctadecanol) to generate biolubricants with improved oxidative stability and low temperature properties. Furthermore, tribological performance of these biolubricants was investigated using four-ball friction and wear tests. The experimental results showed that modified WCO exhibited favorable physicochemical properties and tribological performance, which makes them suitable candidates to formulate eco-friendly lubricants [98]. It is worthwhile to note that the compositional consistency of WCO is a concern, as it depends on the quality of oil used. Moreover, its upgradation by hydrogenation may be required to reduce unsaturation, convert oxygenated functional groups such as aldehydes, ketones and carboxylic acids, and thus improve its storage and thermal stability. Jahromi et al. [20] modified fatty acids and WCO through a series of chemical steps involving hydrolysis, dehydration/ketonization, Friedel–Crafts acylation/alkylation and mild hydrotreatment to produce biolubricants containing saturated linear hydrocarbon chains with cyclic rings and polar moieties in the structure. Importantly, they used model compounds such as anisole, 2-methyl furan, cyclopentanol, and cyclopentanone, which are typical pyrolysates from biomass, in the condensation step to produce the desired molecules. The biolubricants had a pour point of –12 °C, a viscosity of 47.5 cP (at 40 °C), a viscosity index of 186, and a total acid number lesser than 1 $mg_{KOH}/g$ [20].

The need of vegetable oil for cooking and the resultant costs make this oil less suitable for lubrication applications. In this context, it becomes imperative to explore the use of alternate waste-derived lubricants such as pyrolysis oil and HTL oil for lubrication applications. The viscosity (at 50 °C) of pyrolysis bio-oil derived from wood and heavy fuel oil have been presented by researchers to be 40–100 cP and 180 cP, respectively. The specific gravity values were 1.2 and 0.94 [76,77]. Similarly, for two samples of HTL oil, the density at 295 K was $1.14 \pm 0.02$ kg m$^{-3}$, and the viscosity at 313 K (mPa s) were 67,000 ($\pm$5000), 2200 ($\pm$200). Viscosity at 353 K (mPa s) were 520 ($\pm$40) and 210 ($\pm$10), respectively [99]. Pyrolysis bio-oil and HTL bio-crude also require mild upgradation via hydrodeoxygenation in order to reduce the oxygen content to acceptable levels. While the presence of oxygen in the biolubricant base stock may be necessary for facile bio-degradation, it eventually reduces the calorific value of the lubricant and, based on the nature of oxygenated functionality, can also adversely affect the storage stability of the oils. The presence of multiple oxygenated functionalities in a molecule can lead to high acidity, measured by total acid number, and an increase in molecular weight due to autoxidation and polymerization reactions. While HTL bio-crudes can be used in applications that demand high viscosity, the excessively high viscosity of HTL bio-crude can affect the flowability of the biolubricant, which is a concern. Incorporating additives such as alcohols in small quantities is shown to improve the storage stability of the biomass-derived pyrolysis oils. Therefore, the additions of antioxidants and thinners are essential for employing HTL bio-crudes as base stocks for biolubricants.

The required properties of cutting fluids are as follows: (a) excellent lubricating properties; (b) suitable viscosity; (c) cheap cost; (d) good cooling capability; (e) non-corrosive; (f) high flash point; (g) chemically stable; (h) low evaporation rate; and (i) non-allergic [100]. Based on the lubricant properties required, waste-derived oils, waste animal fats, greases and tallow can be sustainable, eco-friendly and cheap, meeting the needs of biolubricants.

**Table 4.** Advantages and disadvantages of biolubricants [15,101].

| Sl. No. | Advantages | Disadvantages |
| --- | --- | --- |
| 1 | High lubricity | High cost. |
| 2 | High viscosity index | Several vegetable oils are edible. This can lead to food vs. fuel debate. |
| 3 | High volatility | Vegetable oils have higher melting points. |

**Table 4.** *Cont.*

| Sl. No. | Advantages | Disadvantages |
|:---:|:---:|:---:|
| 4 | High boiling point (lower emissions) | Vegetable oils have low oxidative stability. |
| 5 | Longer tool life | Biolubricants are less developed compared to fossil-based technologies. |
| 6 | Better skin compatibility | Poor oxidation stability of pyrolysis bio-oils. |
| 7 | Better safety on the shop floor | High acidity of pyrolysis bio-oils. |
| 8 | Biodegradability is high (as they are free of aromatics) | Higher extent of upgradation required for thermochemically derived base stocks. |
| 9 | High volatility | High viscosity of HTL biocrudes. |
| 10 | Customizable chemical structures | |
| 11 | Lesser amount of contaminants | |
| 12 | The base stocks for biolubricants can be derived from a variety of sources | |

## 6. Applications

Table 5 presents some oils derived from renewable feedstocks and their major applications. Engine oils tend to minimize the transport of contaminants and other particulates, and keep them away from the moving parts [102]. As engine oils undergo oxidative degradation and wear during service, it is vital to characterize the ageing of engine oils at simulated conditions to understand and evaluate the performance of existing oils and also design new formulations [52]. Researchers have employed pongamia oil as a compression ignition (CI) engine lubricant [83], showing that it improves efficiency and, unlike a mineral oil lubricant, can potentially eliminate the emission of trace metals, as it is devoid of any metal constituents.

Cutting fluids control the temperature rise by giving adequate lubrication and cooling between the workpiece and the tool [103]. Even though mineral oils are comparatively cheaper for this application, they exhibit poor performance due to low-temperature solidification, oxidative instability, and loss of viscosity at elevated temperatures. They are also susceptible to explosion in the presence of an oxidizing agent. Importantly, the additives used to enhance the performance of the lubricant may be dangerous to humans and the environment. Non-edible vegetable oils are biodegradable, and are ultimately decomposed and mineralized into carbon dioxide and hydrogen by microbes. Biodegradability ensures the safe integration of biomaterial back into the carbon cycle of nature. Non-edible vegetable oils degrade faster than mineral oils in the natural environment [103,104].

Kania et al. [105] presented a review of biolubricants in drilling fluids. A good drilling fluid exhibits these properties: relatively high viscosity, low corrosivity, high lubricating film strength, low flammability, high solubility, low pour point, high thermal and oxidative stability, and non-toxic. Lubricants are majorly applied to water-based mud (WBM) in drilling applications as the lubricity is inadequate. A biolubricant must possess favorable lubricity, solvency, viscosity, VI, thermal-oxidative and hydrolytic stability in order to exhibit excellent lubricating performance under wellbore conditions. These properties are influenced by the presence of an ester functional group. Most esters used as lubricants in drilling fluids are derived from polyhydric alcohols. Only some esters are derived from monohydric alcohols, including aliphatic esters and diesters [105].

The minimum quantity lubrication (MQL) is an environmental protection technology that employs a nozzle to spray a small amount of lubricating fluid and compressed gas into the cutting zone for the purpose of cooling and lubrication [106,107]. Dong et al. [107] discussed the temperature of the MQL milling of the 45 steel using cottonseed, palm, castor, soybean, and peanut oils as base stocks. The effects of the carbon chain length, thermal conductivity, fatty acid composition and viscosity on the milling temperature were studied. By simulating the temperature distribution of the milling of the 45 steel with five different

vegetable oils, it was shown that the cottonseed and the palm oils exhibited a good cooling effect. This is due to the presence of short carbon chain length palmitic acid in cottonseed and palm oils, which is conducive to the MQL milling.

Gear oils are necessary for industrial and automotive lubrication, wherein they are generally used in differentials, transmissions, power take-offs and non-drive applications. Besides the usual functions of lubricants, these oils are required to reduce noise, inhibit corrosion, transfer heat, and improve overall efficiency [108,109]. In order to obtain good protection in both the hydrodynamic and elastohydrodynamic regimes, strict viscosity regulation of the base oil is needed. While vegetable oils do not possess the required viscosity for these applications, this limitation can be overcome by thermal polymerisation of the oil by heating the oil in inert ambience to generate high molecular weight products [109]. For high viscosity applications, lubricants of the required quality can be generated from wastes, and biocrudes derived through hydrothermal liquefaction can be suitably tailor-made to meet the requirements. This is also evident from the viscosity values of these fuels shown in Table 3.

**Table 5.** Oils, their properties, and their major applications.

| Sl. No. | Oil | Major Properties * | Major Applications | Reference |
|:---:|:---:|:---:|:---:|:---:|
| 1 | Palm oil | Less corrosive, low coefficient of friction, high viscosity | Greases, metal working fluids (MWFs) | [2,15] |
| 2 | Coconut oil | High antiwear, better lubricity, low coefficient of friction | Engine oils | [2,15] |
| 3 | Crambe oil | n.a. | Greases, surfactants, cosmetics, chemicals | [15] |
| 4 | Sunflower oil | | Diesel fuels, greases | [2,15] |
| 5 | Soybean oil | | Hydraulic oils, biodiesel fuel, engine oils, transmission fluids, printing inks, paints, detergents, coatings, pesticides, shampoos | [2,15] |
| 6 | Safflower oil | High VI, high flash point than some conventional oils, high lubricity, low evaporative loss, low co-efficient of friction, better lubricity, non-toxic | Resins, diesel fuels, enamels | [2,15] |
| 7 | Linseed oil | | Stains, coatings, vanishes, paints | [2,15] |
| 8 | Olive oil | | Engine oils | [2,15] |
| 9 | Canola oil | | MWFs, transmission fluids, food-grade lubes, hydraulic fluids, penetrating oils, transmission fluids | [2,15] |
| 10 | Castor oil | High VI, low volatility, high antioxidants, low deposit formation | Greases, gear lubricants | [2,15] |
| 11 | Pongamia oil | Low frictional losses, low emissions, minimum break-specific fuel consumption and high break thermal efficiency at medium loads | Power transformer applications, anticorrosive coating | [2,110,111] |
| 12 | Tallow oil | n.a. | Soaps, cosmetics, plastics, hydraulic oils | [15] |
| 13 | Cuphea oil | n.a. | Motor oils, cosmetics | [15] |
| 14 | Jojoba oil | n.a. | Greases, cosmetics, lubricants | [15] |
| 15 | Jatropha oil | High VI, low wear loss, low cumulative weight loss, low coefficient of friction | Biodiesel | [2,112,113] |

**Table 5.** *Cont.*

| Sl. No. | Oil | Major Properties * | Major Applications | Reference |
|:---:|:---:|:---:|:---:|:---:|
| 16 | Rapeseed oil | Better oxidation stability, better cold flow property and low coefficient of friction | Power transformer applications, hydraulic fluids, greases, chainsaw oils | [2,15,110] |
| 17 | HTL liquid | High viscosity, high acidity | Biocrude for various engineering applications, heating, in marine, rail engines, can be upgraded to transportation fuels and jet fuels | [95,114] |
| 18 | Pyrolysis oil | High viscosity if the feedstock is polymers, while viscosity is low if derived from biomass; poor oxidation stability, high acidity | Biofuel for energy applications, heating, steel, cement industries, generating valuable hydrocarbons and petrochemicals | [81,86,88] |

n.a.—Not available; * the reference fuels used for the comparison of the oil properties include mineral oil, SAE20W40, SAE 40, and SAE20W50, as elucidated in the references.

Technoeconomic analyses and life cycle assessment of biolubricants involving cradle-to-grave and cradle-to-wheel approaches must be conducted to understand the commercial viability of biolubricants vis-à-vis conventional mineral oil-based lubricants. Importantly, such studies are scarce in the literature. Athaley et al. [115] performed a technoeconomic analysis of butadiene, jet fuel, surfactant, and lubricant production from furfural, which is a biomass-derived platform molecule. A sequential catalytic process involving the acylation of anhydride and furan, the hydrogenation of 2-pentanoyl furan, the hydroalkylation of pentyl furan, and the hydrodeoxygenation of condensed furan was analyzed. It was found that the raw material cost was nearly 87% of the total operating cost of the process, and the minimum selling price of the lubricant was USD 4037 per metric ton [115]. This is due to the high price of raw materials such as valeric acid, lauraldehyde, and lauric acid. This necessitates the integration of biolubricant production processes within biorefineries. Moreover, a thorough study of different scenarios of biolubricant production in typical bioethanol, biodiesel, oleochemical, and waste biorefineries is required.

## 7. Conclusions

In this review, the major advantages, disadvantages and applications of edible, non-edible and mineral oil-based lubricants have been discussed. While vegetable oil-based lubricants are popular owing to the good lubricative properties they offer, along with low toxicity, eco-friendliness and biodegradability, their requirement for cooking food can escalate cost, and affect their availability at large scales. Non-edible oils also present an interesting option from the viewpoint of waste management and value generation from wastes. In this context, the use of bio-oils generated from lignocellulosic agro-residues, microalgae, and plastic-rich fractions such as refuse-derived fuels through thermochemical technologies such as pyrolysis and HTL need clearer focus.

From the literature, it can be noted that biolubricants can provide better lubrication properties than the conventionally used synthetic and mineral oils, and more importantly, the essential properties required for diverse engineering applications can be obtained. Additionally, as they are non-toxic and clean, biolubricants may potentially be utilised in highly sensitive applications such as marine and forestry ecosystems, where strict laws, rules and regulations apply. In this context, the demand for biolubricants is expected to only increase in the near future in applications such as hydraulic oils, engine oils, gear oils, cutting fluids, and in electrical appliances and turbomachinery.

The key challenges involved and the scope for further investigation in the area of biolubricants from wastes can be summarized as follows. The properties and composition of thermochemically derived bio-oil or bio-crude are heavily dependent on the type of feedstock, and so, it is imperative to produce biolubricant base stocks from a specific variety of biomass residue or plastic. Any variation of feedstock quality can lead to adverse changes

in properties. In fact, utilizing agro-residues, plastics and their mixtures for biolubricant preparation is a sustainable option, and is a high-value proposition for waste as compared to its regular utilization as transportation-grade or stationary engine fuel. Systematic studies on novel biolubricant formulations using bio-oil/bio-crude along with additives need to be performed to expand our current understanding of the physicochemical, thermophysical, tribological properties and stability of the biolubricants. Suitable waste feedstocks used for the preparation of such biolubricants include biomass, and olefinic plastic mixtures, waste cooking oil, and catalytically upgraded pyrolysis oils derived from polymers. Usually, pyrolysis oils derived from polymers without catalytic treatment exhibit a waxy nature and present high viscosity values. Therefore, catalytic treatment or secondary pyrolysis treatment to improve the flowability of the base oil is a necessity.

Mild to moderate upgradation of oils from agroresidues may be necessary to reduce the oxygen content in the fuel. This is usually performed using catalysts in hydrogen ambience. While this is a well-known process, controlled hydrogenation and hydrodeoxygenation are necessary to tailor the properties of biolubricants. Studies on the stability of both existing vegetable oil-based biolubricants and waste-derived biolubricants are scarce. A basic investigation of storage stability, oxidative stability, hydrolytic stability and thermal stability is required in accordance with pre-existing standards. A thorough assessment of biodegradability and ecotoxicity of the biolubricant formulations needs to be performed, and the existing literature must be expanded. Detailed comparative studies to investigate the advantages and disadvantages of various biolubricants for diverse applications need better focus. The limited studies available indicate that the high price of raw materials can make the biolubricant production process unsustainable from a monetary viewpoint. This requires a thorough technoeconomic assessment of using vegetable oil-based and waste-derived biolubricants in a closed-loop biorefinery for a successful large-scale transition to biolubricants. In addition, impact metrics such as life cycle $CO_2$ emissions that directly affect climate change, water depletion, fossil depletion, and land-use change need to be evaluated based on thorough life cycle analysis. Key policy decisions are determined by the energy and economic benefits offered by the biolubricants over conventional petroleum-derived lubricants. Sometimes, even the blending of biolubricants with conventional lubricants may be a sustainable option from the viewpoint of biodegradability, although this is yet to be fully explored. The authors are confident that biolubricants are a promising and potential option to utilize waste and generate value while achieving a sustainable and circular bioeconomy.

**Author Contributions:** Conceptualization, methodology, validation, formal analysis, data curation, writing—original draft preparation, R.N.S.; conceptualization, methodology, formal analysis, writing—original draft preparation, resources, supervision, project administration, funding acquisition, R.V. All authors have read and agreed to the published version of the manuscript.

**Funding:** The authors thank Department of Science and Technology (DST), India, for funding the study through Waste Management Technologies Grant # DST/TDT/WMT/AgWaste/2021/15.

**Institutional Review Board Statement:** Not applicable.

**Informed Consent Statement:** Not applicable.

**Data Availability Statement:** Not applicable.

**Acknowledgments:** R.N.S. thanks the DST, India for funding the research associateship. The National Center for Combustion Research and Development is sponsored by Department of Science and Technology, India.

**Conflicts of Interest:** The authors declare no conflict of interest. The funders had no role in the design of the study; in the collection, analyses, or interpretation of data; in the writing of the manuscript, or in the decision to publish the results.

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
