# Peer review of "Current Status and Future Prospects of Biolubricants: Properties and Applications"

_lubricants, doi:10.3390/lubricants10040070_

Round 1

Reviewer 1 Report

This manuscript provided a brief review for the properties and applications biolubricants. However, it is not suitable for publication in Lubricants.

  1. The organization of the introduction as a whole is disordered. There is no clear theme for each paragraph and there is a lack of relation between sentences.
  2. Some literature should not be cited as they are unrelated with biolubricants, e.g. ref. 20, 24, 25. Furthermore, the description on ref. 20, 24, 25 is not logical as well.
  3. There are many unreasonable descriptions in the whole manuscript. For example, the introduction of additive package for lubricants at the beginning of section 2. is not related with the following content, as well as the induction of different lubricant additives (line. 138-146).
  4. Please check the data of kinematic viscosity at 313K, as they are too low for vegetable oils. Although there is a big gap between biolubricants (especially non-edible or edible oils) and synthetic/mineral oils, the authors mentioned that biolubricants generally have high viscosity (line 208).
  5. The sections 3.2-3.5 lack the data and corresponding literature of biolubricants, and the ref. 52 is not related with biolubricants. Table 4 is also unrelated with the content of section 3.5.
  6. The flask point of vegetable oil is inconsistent with table 3 (line 412).
  7. Pyrolysis and HTL oils are mostly used as biofuels, and studies on their application as lubricants are also needed to be cited.
  8. There are many grammatical and spelling mistakes, e.g. Can be naphthenic, aromatic, or paraffinic in nature. Mainly include API Group I, II, III oils. (Table 1). Higher the VI, lower is the change in viscosity of the lubricant… (line 215).

Author Response

Please find the responses attached

Reviewer 2 Report

TITLE

The title of the manuscript adequately describes the information presented in this manuscript.

Abstract

The abstract of this manuscript is clear and concrete on the theme developed with the information that has been compiled and analyzed for this reason.

Introduction

This section is adequately supported by the references that support the information analyzed and presented in this manuscript. But it can be improved by presenting updated references less than 5 years old at the time of this writing.

SECTIONS 2 TO 6

The sections presented in this manuscript and listed below are framed within some of the points of interest on the development of biolubricants.

  1. Sources and preparation of lubricating oils
  2. Properties of Biolubricants4. Development of biolubricants from biomass via thermochemical techniques
  3. Comparison of biolubricants with other lubricants and the need for alternate biolubricants
  4. Applications

But this perspective has some shortcomings, such as a technical and economic analysis of the production of biolubricants. This last perspective would be of great interest to the scientific community and the general public.

Conclusion

The conclusions of the article are clear and concrete. They present an overview of the prospects for biolubricants, focusing on their advantages and disadvantages. But as part of these perspectives, the techno-economic analysis of industrial scaling for the production of these biolubricants is needed. It is important to consider within the manuscript current perspectives such as current studies on large-scale production of biolubricants and improvements in production processes to make them economically profitable.

References

The 39.45% percent are references from the period 2017-2022, 26.34% from the period 2011-2016, 34.21% are references are are lower than the year 2010. In general, 60.55% of the references it is older than 5 years.

An effort has been made to update the references that support the research. Even so, it is recommended to have a greater number of references no greater than 5 years from the date the manuscript is submitted for review. The updated references allow us to observe the trends in the area of ​​biolubricants and the novelty of the information analyzed and described in the manuscript.

Author Response

Please find the responses attached

Reviewer 3 Report

The mini-review is written well explaining on the fundamentals of the bio-lubricants. I have few comments before it can be published.

  1. Line 229: it is sulfur not dulfur; line 270: performed, not perfoemd
  2. No studies reported on bio-lubricants production from pyrolysis and hydrotheml liquefaction. If there are, please report and discuss in detail. Accordingly, revise the section 4.
  3. It would be more meaningful if the authors compare the properties of some conventional and biolubricant properties from various sources in section 5.
  4. Lines 446 - 448, as there are no studies reported, authors need to check these sentences.
  5. Lines 454 - 456, in addition to the waste derived oils, waste animal fats, geases and tallow could be used as a potential eco-friendly alternatives for conventional lubricants.

Author Response

Please find the responses attached

Reviewer 4 Report

The Authors have presented the review pointing out the main 
features of the existing biolubricants, and puts forward the case of using sustainable biolubricants,  which can be generated from agro-residues via thermochemical processes. The topic is interesting and can be considered for publication after incorporating following suggestions:

  1. Please highlight the key findings of review in the abstract section
  2. In literature section mention the contribution of each cited paper individually rather to mention them accumulatively
  3. Some more literature of biolubricants should be added
  4. The conclusion section should not be the part of future direction 
  5. Source of Table 3 should be added.

Author Response

Please find the responses attached

Round 2

Reviewer 1 Report

The author has added a significant amount of content and detail to the manuscript according to the comments, in addition, the language has also been greatly refined. Overall, the quality of the article is greatly improved compared to the previous manuscript, and I think the current version is acceptable to Lubricant.